# Inhibition of Cancer Cell Migration and Invasion In Vitro by Recombinant Tyrosine-Sulfated Haemathrin, A Thrombin Inhibitor

**DOI:** 10.3390/ijms252111822

**Published:** 2024-11-04

**Authors:** Guk Heui Jo, Sun Ah Jung, Jin Sook Yoon, Joon H. Lee

**Affiliations:** 1Department of Cell Biology, Myung-Gok Eye Research Institute, Konyang University College of Medicine, Daejeon 35365, Republic of Korea; cooki28@kimeye.com (G.H.J.); ayong@hanmail.net (S.A.J.); 2Department of Ophthalmology, Severance Hospital, Institute of Vision Research, Yonsei University College of Medicine, Seoul 03722, Republic of Korea

**Keywords:** tyrosine sulfation, thrombin, madanin-1, cancer, migration, invasion, metastasis, epithelial–mesenchymal transition, sulfated haemathrin

## Abstract

Thrombin, a key enzyme in the regulation of hemostasis, has been implicated in cancer progression. This study explored the effect of recombinant tyrosine-sulfated haemathrin on cancer cell behavior and signaling pathways compared to wild-type (WT) haemathrin 2. The recombinant proteins, tyrosine-sulfated haemathrin 2 (haemathrin 2S), and WT haemathrin 2 were produced in *Escherichia coli* and subsequently purified and applied to SKOV3 and MDA-MB-231 cells with and without thrombin stimulation. Cell migration and invasion were assessed using wound healing and Transwell assays, respectively. Haemathrin 2S treatment significantly diminished cell migration and invasion promoted by thrombin in both SKOV3 and MDA-MB-231 cells (*p* < 0.05). Additionally, haemathrin 2S effectively inhibited thrombin-induced phosphorylation of serine/threonine kinase (Akt) in both cell lines (*p* < 0.05), while WT haemathrin 2 had this effect only in MDA-MB-231 cells. Furthermore, haemathrin 2S significantly reduced thrombin-activated phosphorylation of extracellular signal-regulated kinases (ERK) and p38 in both cell lines (*p* < 0.05) and reversed E/N-cadherin expression in thrombin-treated MDA-MB-231 cells (*p* < 0.05), which were not observed with WT haemathrin 2. Overall, haemathrin 2S was more effective than WT haemathrin 2 in reducing cancer cell migration and invasion, indicating that targeting thrombin with sulfated haemathrin is a promising strategy for cancer therapy. However, further in vivo studies are needed to confirm these results.

## 1. Introduction

Metastasis, which is defined as the spread of cancer cells from the primary tumor to distant organs, is a leading cause of cancer-related mortality. Metastasis involves several key steps, including cell migration, invasion through the extracellular matrix, and subsequent colonization of secondary sites [1,2]. Given the critical role of these mechanisms in tumor progression, there is growing interest in identifying novel therapeutic agents that can effectively inhibit cell migration and invasion, thereby mitigating metastasis [3]. Thrombin, a pleiotropic serine protease involved in the coagulation system, is reportedly associated with cancer cell metastasis [4,5,6], and its expression is significantly elevated in patients with cancer, particularly those with metastasis. Tissue factors initiate thrombin generation, and the elevated thrombin levels in various cancers suggest its involvement in promoting tumor growth, angiogenesis, and metastasis. Thrombin contributes to cancer cell invasion and migration by modulating protease-activated receptors (PARs), especially PAR-1, which trigger a cascade of downstream signaling pathways [7,8]. These pathways include the phosphorylated serine/threonine kinase (Akt) and p38 Mitogen-activated protein kinase (MAPK) pathways, which are pivotal for enhancing cell survival, motility, and invasiveness [9]. Thrombin also induces a prometastatic microenvironment by promoting cancer cell-platelet interaction, shielding circulating tumor cells from immune surveillance, and aiding their dissemination [5,9]. Clinical research indicates that anticoagulants reduce thromboembolic complications linked to cancer, potentially enhancing survival by hindering cancer progression and may also prevent cancer development [10].

Given the multifaceted role of thrombin in promoting cancer metastasis, targeting thrombin or its downstream effects has become an area of significant interest in cancer research. Several anticoagulant substances have been isolated and characterized to date from blood-feeding invertebrates [11,12,13,14]. Additionally, the search for thrombin inhibitors with anti-metastatic potential has led to the exploration of various bioactive compounds [15]. Many thrombin inhibitors identified in ticks belong to the Kunitz-type proteinase inhibitor family. These inhibitors are present in both soft ticks (e.g., monobin from *Argas monolakensis,* ornithodorin from *Ornithodoros moubata*, and savignin from *O. savignyi*) and hard ticks (e.g., boophilin from *Rhipicephalus (Boophilus) microplus*, amblin from *Amblyomma hebraeum*, and hemalin from *Haemaphysalis longicornis*) [16,17,18,19,20,21]. These inhibitors feature a double-domain structure, wherein the N-terminal domain non-canonically inhibits the active site, and the C-terminal domain binds to exosite-I. Despite their high potency and specificity, soft tick thrombin inhibitors function as kinetically slow, tight-binding competitive inhibitors. Hirudin from *Hirudo medicinalis* is one of the most well-known anti-thrombin inhibitors, targeting the enzyme active site and exosite-I, and is currently used in anticoagulant therapy along with its simplified analog, hirulog [14]. Recently, a new class of thrombin inhibitors, called madanins, was discovered and characterized in the salivary glands of *Haemaphysalis longicornis* [13,22,23,24]. These cysteine-free thrombin inhibitors belong to a distinct inhibitor family, classified as family I53 in the MEROPS database. Furthermore, the post-translational modification, tyrosine sulfation, in madanin proteins is critical for potent antithrombotic and anticoagulant activities [25]. The sulfated form of madanin-1 further suppressed migration and invasion of cancer cells in vitro than wild-type (WT) madanin-1 [13], highlighting the ability of madanin-1 to disrupt thrombin-mediated signaling pathways, thereby reducing the metastatic potential of cancer cells.

In line with these findings, we focused on another madanin-like thrombin-targeting agent, haemathrin [22]. Haemathrins are peptide anticoagulants that lack cysteine residues and share approximately 65–70% sequence similarity with madanins. They are classified into the inhibitor I53 superfamily in the MEROPS database. Recombinant haemathrins exhibit anticoagulant properties and selectively inhibit thrombin production [22]. Using a genomic approach, haemathrins 1 and 2 were isolated from the salivary glands of *Haemaphysalis bispinosa*, a hard tick belonging to the Ixodidae family. *H. bispinosa* is the most common tick species of the genus *Haemaphysalis* found in India [26]. In this study, we investigated the effect of recombinant tyrosine-sulfated haemathrin 2 on thrombin-induced migration and invasion of cancer cells and intracellular signaling pathways. We used the SKOV3 ovarian cancer cell line and the MDA-MB-231 breast cancer cell line, which are well-characterized, aggressive, and highly invasive cancer cells, as our primary focus was on assessing thrombin’s impact and the effect of thrombin inhibitor in the aggressive cancer phenotype. This study highlights the potential of haemathrin 2S as a promising candidate for targeting thrombin in cancer therapy by suppressing cancer cell migration and invasion.

## 2. Results

### 2.1. Inhibition of Cancer Cell Migration by Haemathrin 2S

We examined the effect of sulfated haemathrin on the migration of SKOV3 and MDA-MB-231 cells using a wound-healing assay (Figure 1) (Appendix A). Cells were pretreated with 10 μg/mL GST, haemathrin 2 WT, or haemathrin 2S for 30 min before treatment with 2 units/mL thrombin for 24 h. The relative wound area of thrombin-treated SKOV3 cells was significantly lower (23%) than that under serum-free conditions (54% in SKOV3 and 55% in MDA-MB-231 cells). Although not significant, thrombin-induced cell migration in MDA-MB-231 cells led to smaller wound areas (42%) than those under serum-free conditions (55%). Thrombin-induced cell migration was significantly inhibited by haemathrin, and this inhibitory effect was more potent with haemathrin 2S (relative wound area: 63% in SKOV3 and 70% in MDA-MB-231) than with haemathrin 2 WT (relative wound area: 36% in SKOV3 and 50% in MDA-MB-23) in both SKOV and MDA-MB-231 cells. Thrombin-induced cell migration in both cell lines was even lower in haemathrin 2S-treated cells than in serum-free conditions without thrombin stimulation.

### 2.2. Inhibition of Cancer Cell Invasion by Haemathrin 2S

The invasive capacity of cancer cells was analyzed using a Transwell cell invasion assay (Figure 2) (Appendix A). Cells were treated with thrombin (2 units/mL) with or without 10 μg/mL GST, haemathrin 2 WT, or haemathrin 2S for 24 h. The mean density of invaded cells/field was 15 in SKOV3 cells and 13 in MDA-MB-231 cells, which were significantly increased under thrombin stimulation into 118 in SKOV3 cells and 113 in MDA-MB-231 cells. Compared with haemathrin 2 WT, haemathrin 2S significantly suppressed thrombin-induced cell invasion in SKOV3 cells and MDA-MB-231 cells by 182% and 178%, respectively.

### 2.3. Effect of Haemathrin WT and Haemathrin 2S on Thrombin-Associated Signaling Pathway

Western blot analysis was performed to investigate the potential mechanisms of action of WT and 2S haemathrin on thrombin-associated signaling pathways (Figure 3). Cells were pretreated with 10 μg/mL GST, haemathrin WT, or haemathrin 2S for 30 min, followed by treatment with thrombin (2 units/mL for 15 min). The phosphorylation of Akt, extracellular signal-regulated kinase (ERK), and p38 was induced by thrombin in both cancer cell lines. Haemathrin 2S significantly inhibited the thrombin-induced phosphorylation of Akt, ERK, and p38 in SKOV3 cells, whereas haemathrin WT did not show any inhibitory effect. In MDA-MB-231 cells, both haemathrin 2S and WT significantly suppressed thrombin-induced Akt phosphorylation, but phosphorylation of ERK and p38 activation by thrombin was significantly inhibited by haemathrin 2S treatment, but not by haemathrin 2 WT.

### 2.4. Effects of Haemathrin WT and Haemathrin 2S on Thrombin-Related E- and N-Cadherin Protein Expressions

We analyzed protein expression, including that of cadherin, in MDA-MB-231 cells using western blotting (Figure 4). Cells were pretreated with 10 μg/mL GST, haemathrin WT or haemathrin 2S for 30 min, followed by treatment with thrombin (2 units/mL for 24 h). Thrombin suppressed E-cadherin expression, which was significantly increased by haemathrin 2S but not by haemathrin WT. Furthermore, thrombin-induced N-cadherin expression was significantly suppressed by haemathrin 2S but not by haemathrin WT.

## 3. Discussion

In this study, we explored the effects of tyrosine-sulfated haemathrin, haemathrin 2S, on cancer cell behavior in comparison to haemathrin 2 WT, using SKOV3 and MDA-MB-231 cell lines. Our findings indicated that haemathrin 2S significantly impeded thrombin-induced migration and invasion in both cell lines, which was not observed with WT haemathrin. These results highlight the potential of haemathrin 2S as an effective inhibitor of thrombin-mediated cancer progression.

Given that the procoagulatory serine protease thrombin promotes tumorigenesis, its role in cancer metastasis has been increasingly recognized, particularly its ability to activate signaling pathways such as PI3K/Akt and MAPK/ERK through PAR-1 receptors, which enhance cancer cell migration, invasion, and survival [27,28,29]. Thrombin can induce the expression of PAR-1, which is highly expressed in highly metastatic tumors [30]. Thrombin also induced spindle-shaped morphological alterations in SKOV3 cells, which may be conducive to invasion and metastasis [31]. The impact of thrombin on the tumor microenvironment, including its interaction with cancer cells and platelets, further contributes to tumor progression and metastasis. [4]. Inflammatory responses triggered by thrombin may also lead to increased expression of cytokines, adhesion molecules, angiogenic factors, and matrix-degrading proteases that promote tumor cell proliferation, angiogenesis, invasion, and metastasis [5]. In the present study, the significant reduction in Akt, ERK, and p-38 phosphorylation in both cancer cell lines treated with haemathrin 2S, as opposed to the limited effect of WT haemathrin, suggests that haemathrin 2S effectively disrupts thrombin-induced signaling cascades. Future studies will include experiments with the specific kinase inhibitor of signal molecules to further understand how haemathrin 2S functions in relation to these pathways.

Moreover, haemathrin 2S reversed the expression of E- and N-cadherins in thrombin-treated MDA-MB-231 cells, indicating its potential role in modulating epithelial–mesenchymal transition (EMT), a critical process in cancer metastasis. During EMT, epithelial cells undergo transcriptional changes, leading to reduced cell adhesion and increased capacity for migration or invasion [32]. EMT is characterized by the downregulation of E-cadherin, a tumor suppressor protein, and the upregulation of N-cadherin, which is associated with increased invasiveness and poor patient prognosis [33]. A reduced E-cadherin expression is a characteristic of advanced carcinoma, in contrast to the upregulation of N-cadherin. N-cadherin interacts with fibroblast growth factor receptor 1, leading to the continuous activation of the MAPK-ERK pathway and the promotion of tumor cell survival via stromal interaction [34]. Thrombin treatment can induce a morphological change from a rounded or epithelial-like shape to a spindle-like phenotype accompanied by a decrease in E-cadherin levels and the activation of PAR-1, which promotes migratory effects [29]. In our previous study, sulfated madanin-1, a thrombin inhibitor, significantly repressed the thrombin-induced switch from E-cadherin to N-cadherin, indicating that madanin-1 2S could have a therapeutic effect on cancer cell metastasis [13]. In the present study, the ability of haemathrin 2S to restore E-cadherin expression while simultaneously reducing N-cadherin levels suggests its potential to inhibit metastasis by interfering with thrombin-induced EMT.

Sulfation is crucial in modulating the therapeutic efficacy of proteins by influencing their biochemical properties. The addition of sulfate groups can enhance the binding affinity of a protein to its target receptors, often leading to increased potency [35]. This post-translational modification also contributes to protein structure stabilization by preventing enzymatic degradation and extending its half-life within the body [36]. Moreover, sulfation improves protein solubility, ensuring better bioavailability and more efficient distribution to target tissues [37]. Tyrosine sulfation of thrombin inhibitors also enhances their anticoagulant activity [38]. For example, sulfation of a tyrosine residue within the acidic tail of the anticoagulant hirudin enhances the affinity of hirudin for thrombin by more than 10-fold [38]. Disulfated madanin-1 prolonged thrombin time to a similar degree at 500-fold lower concentrations as did unsulfated madanin-1 [25]. The findings of this study, using haemathrin 2S in comparison with haemathrin WT, are consistent with those observed in studies on madanin-1 2S, which also demonstrated a significant reduction in cancer cell migration, invasion, and key signaling pathways following sulfation [13]. Thus, sulfation significantly enhances the therapeutic potential of proteins, making it a valuable strategy for drug development.

## 4. Materials and Methods

### 4.1. Expression and Purification of Haemathrin 2 WT and Sulfation Proteins

The haemathrin 2 WT was purchased from Cosmo Genetech (Seoul, Republic of Korea). The two tyrosine codons of WT haemathrin 2 were changed to two TAG for haemathrin 2 sulfation (haemathrin 2S) (Table 1). Nucleotides were added to introduce the *EcoRV* and *XhoI* restriction sites. The synthesized double-stranded oligonucleotide was inserted into the *EcoRV-* and *XhoI*-digested pET-41a (Novagen) vector, which contains an N-terminal GST, polyhistidine (6xHis) tag, and C-terminal polyhistidine for purification (Figure 5A).

To co-transform haemathrin 2 WT pET-41a (50 μg/mL kanamycin) and haemathrin 2S pET-41a and pSUPAR6-L3-3SY (50 μg/mL kanamycin and 50 μg/mL chloramphenicol) in BL21, this mixture is heat-shocked (usually at 42 °C for 2 min) to facilitate the cells’ absorption of DNA. Then, the mixture is incubated in a shaking incubator in 1 mL of LB medium for 1 h without antibiotics. After incubation, the LB mixture is spread over LB agar plates (50 μg/mL kanamycin and 50 μg/mL chloramphenicol) and incubated for 24 h in an incubator at 37 °C. The next day, we check the colony and incubate it again in LB broth to see if it contains plasmid [39]. The positive BL21 colonies were selected and cultured overnight in 3 mL of Luria–Bertani (LB) medium. The next day, the cultured *E. coli* strains expressing haemathrin 2 WT were transferred to 250 mL LB medium until OD_600_ = 0.5; then, 0.1 mM IPTG was added and subsequently incubated for an additional 5 h at 37 °C to overexpress the fusion proteins. Cultured *E. coli* strains containing haemathrin 2S were inoculated into 250 mL of LB medium with 10 mM sulfotyrosine (Bachem, Bubendorf, Switzerland) and cultured until OD_600_ = 1; and then, 1 mM IPTG was added and incubated for 20 h at 25 °C to overexpress the haemathrin 2 sulfated proteins (Figure 5B).

The medium was then centrifuged at 10,000× *g* for 10 min, and the bacterial pellets containing proteins were suspended in 10 mL of binding buffer (5 mM imidazole, 0.5 M NaCl, and 20 mM Tris-HCl, pH 7.9) and sonicated. The disrupted cells were centrifuged at 14,000× *g* for 30 min, and the supernatant containing haemathrin 2S and cell pellet containing haemathrin 2 WT were used for further purification. The supernatant containing haemathrin 2S was loaded onto a pre-equilibrated Ni-NTA resin (Novagen, Beijing, China) with distilled water and binding buffer, allowing it to flow through under gravity. The column was washed with 10 volumes of the binding and elution buffers (100 mM imidazole, 0.5 M NaCl, and 20 mM Tris-HCl, pH 7.9). Dialysis with buffer (20 mM Tris-HCl, 50 mM NaCl, and 0.5 mM β- mercaptoethanol, pH 7.5) was performed to remove imidazole, ensuring proper storage of the purified proteins.

A pellet containing haemathrin 2 WT protein was used to isolate inclusion bodies (IBs). For washing, the insoluble pellet was resuspended in IB wash buffer (50 mM Tris-HCl pH 8.0, 2% Triton X-100, and 2% sodium deoxycholate) and centrifuged for 15 min at 10,000× *g*, 4 °C. Then, the resulting pellets (IBs) were resuspended in IB resuspending buffer (50 mM Tris-HCl pH 8.5, 6 M urea, and 150 mM β-mercaptoethanol) and mixed vigorously and incubated overnight at 4 °C. The following day, the dissolved proteins were centrifuged for 30 min at 10,000× *g*, 4 °C, and the supernatant containing solubilized IBs was collected in a tube. The supernatant containing haemathrin 2 WT was loaded onto a pre-equilibrated Ni-NTA resin (Novagen) with distilled water and binding buffer containing 6 M urea and then allowed to flow through the resin under gravity. The column was washed with 10 volumes of binding buffer containing 6 M urea, and haemathrin 2 WT proteins were eluted with the elution buffer (100 mM imidazole, 0.5 M NaCl, 20 mM Tris-HCl, and 6 M urea, pH 7.9). The eluted proteins were then dialyzed overnight at 4 °C using a pump (GE Peristaltic Pump P-1, Amersham Biosciences, Buckinghamshire, United Kingdom). Finally, haemathrin 2 WT and haemathrin 2S proteins were resolved using 10% sodium dodecyl sulfate–polyacrylamide gel electrophoresis (SDS-PAGE). After electrophoresis, the gels were stained with Coomassie Blue R-250 and destained (Figure 5C). The concentration of 10 μg/mL was determined based on the experiment in the previous paper [13,40], and the proteins were dissolved in 20 mM Tris-HCl (pH 7.5) and 50 mM NaCl.

### 4.2. Wound Healing Assay

SKOV3 (2 × 10^5^ cells/well) and MDA-MB-231 (3 × 10^5^ cells/well) were seeded into 24-well plates. After reaching 100% confluence, the cell monolayers were scratched with a 200 µL sterile pipette tip to create a vertical wound. To avoid any influence from the rate of cell growth, the cell culture medium was changed from RPMI-1640 with 10% fetal bovine serum to serum-free RPMI-1640. The cells were incubated with or without 2 units/mL thrombin and with 10 μg/mL of either haemathrin 2 WT or haemathrin 2S. Phase-contrast images were acquired after scratching and after 24 h of incubation at 37 °C. To quantify the effects of the reagents on the scratch-wound area, the migration distance (mm) between the edges of the gap was measured for each image. Wound area measurements and cell migration rate calculations were performed using the ImageJ software (version 6.0; National Institutes of Health), with each experiment performed in triplicate. The relative wound area was calculated using the following formula:Relative wound area (%) = 100 × (24 h area of wound/initial area of the wound).

### 4.3. Invasion Assay

The invasion assay was carried out using an 8 μM pore 24-well Transwell (Costar, Washington, DC, USA) coated with PBS buffer containing 25 µg Matrigel (Sigma-Aldrich, St. Louis, MO, USA) and 0.1% gelatin (Sigma-Aldrich). Cells were grown to 80% confluence in growth medium, followed by synchronization through serum starvation in RPMI-1640 for 24 h. The cells were then seeded (1 × 10^5^ cells/mL) on the top of the chamber [41]. The bottom chamber contained serum-free medium with or without 2 units/mL thrombin in the presence of 10 µg/mL GST, haemathrin 2 WT, or haemathrin 2S. The cells were fixed in 100% methanol for 10 min and stained with 0.1% crystal violet (Thermo-Fisher Scientific, Waltham, MA, USA) for 10 min. The cells on the top side of the filter were removed using a cotton swab. The cells that migrated to the bottom of the filter were micrographed using an Olympus DP70 microscope (Tokyo, Japan), and three fields per sample were captured at 10× magnification. Cell quantitation was performed by measuring the pixel density of the crystal violet-stained cells using a DP controller (Tokyo, Japan).

### 4.4. Western Blotting

Cells were lysed in cold 1X cell lysis buffer (Cell Signaling, Danvers, MA, USA) containing 1 mM phenylmethylsulfonyl fluoride and 50 mM NaF mixture. Protein concentration was determined using a BCA protein assay kit (Thermo Fisher Scientific). Equal amounts (30 µg/lane) of protein were resolved by 6–10% SDS-PAGE. The proteins were transferred to nitrocellulose membranes (Millipore, Burlington, MA, USA) and probed with antibodies (1:1000) against p-Akt (4060), Akt (9272), p-ERk (sc-7383), ERk (sc-1647), p-p38 (9211), p38 (9212), β-actin (4967), E-cadherin (5296) (Cell Signaling), N-cadherin (sc-393933, Santa Cruz Biotechnology, Dallas, TX, USA), and His-probe (sc-803, Santa Cruz Biotechnology). Specific immunoreactions were detected using mouse or rabbit secondary antibodies conjugated to a chemiluminescence reagent (Santa Cruz Biotechnology).

### 4.5. Statistical Analysis

Statistical analysis was conducted using data expressed as the mean ± standard deviation from a minimum of three independent experiments. One-way analysis of variance was used to determine significant differences among the experimental groups. For pairwise comparisons between conditions, Tukey’s post hoc test was applied. Data analysis was performed using IBM SPSS Statistics (version 20.0; IBM Corp., Armonk, NY, USA). Statistical significance was set at *p* < 0.05.

## 5. Conclusions

Our findings suggest that haemathrin 2S is a promising candidate for targeting thrombin in cancer therapy on account of its ability to inhibit key signaling pathways and reverse EMT, ultimately suppressing cancer cell migration and invasion. This highlights the importance of tyrosine sulfation in enhancing the anti-thrombin and anti-metastatic properties of these proteins. While the current study provides promising evidence for the anti-metastatic potential of haemathrin 2S, further research is required to elucidate its precise mechanism of action and evaluate its effectiveness in vivo. Future studies are needed to confirm these findings and to assess the clinical applicability of haemathrin 2S as a novel therapeutic agent for cancer treatment. In addition, exploring the therapeutic potential of 2S in different cancer types in combination with other treatments may offer valuable insights into its role in cancer therapy.

## Figures and Tables

**Figure 1 ijms-25-11822-f001:**
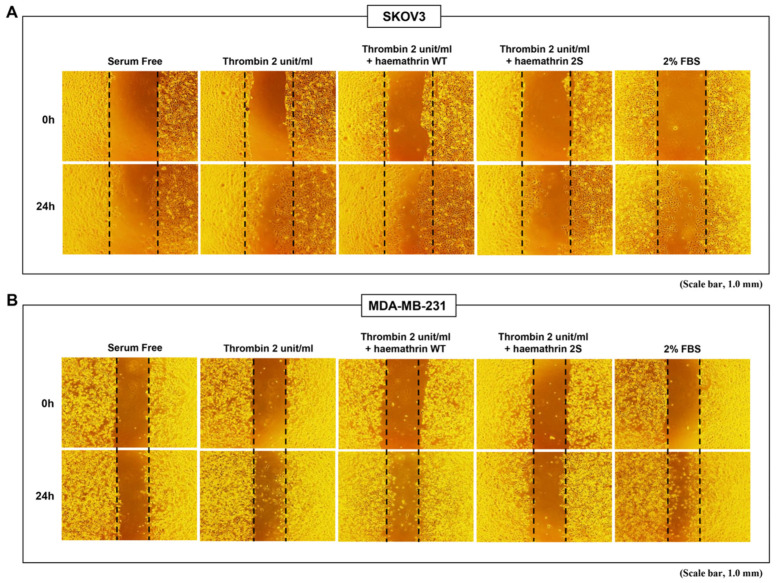
Effect of haemathrin WT and haemathrin 2S on cancer cell migration. A monolayer of confluent cells was scraped with a sterile pipette tip after being preincubated with 10 μg/mL GST, haemathrin WT, or haemathrin 2S for 30 min. (**A**) SKOV3 and (**B**) MDA-MB-231 cells were treated with or without stimulation of thrombin (2 units/mL) for 24 h, and wound closure was observed using phase contrast microscopy was photographed. The average distance between the edges of the gap was measured in three independent experiments. Haemathrin 2S significantly inhibited thrombin-induced migration more effectively than haemathrin WT. Haemathrin WT, haemathrin wild-type proteins; haemathrin 2S, haemathrin 2 sulfation proteins.

**Figure 2 ijms-25-11822-f002:**
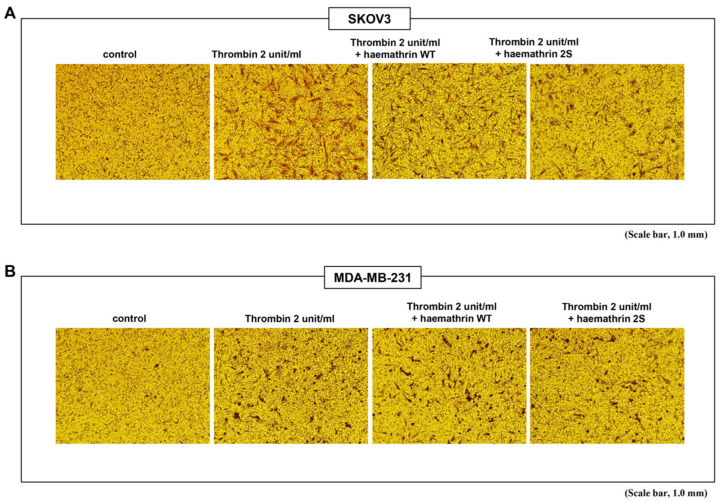
Effect of haemathrin WT and haemathrin 2S on Transwell invasion of cancer cells. Invasion of (**A**) SKOV3 and (**B**) MDA-MB-231 cells was investigated via a Transwell invasion assay with Matrigel for 24 h. The bottom chamber contained serum-free medium with GST, haemathrin WT, or haemathrin 2S (10 μg/mL) in the presence of thrombin (2 units/mL). Representative images indicating the density of the invaded cells per field 24 h after seeding are shown. Only haemathrin 2S treatment significantly inhibited thrombin-induced cell invasion. Haemathrin WT, haemathrin wild-type proteins; haemathrin 2S, haemathrin 2 sulfation proteins.

**Figure 3 ijms-25-11822-f003:**
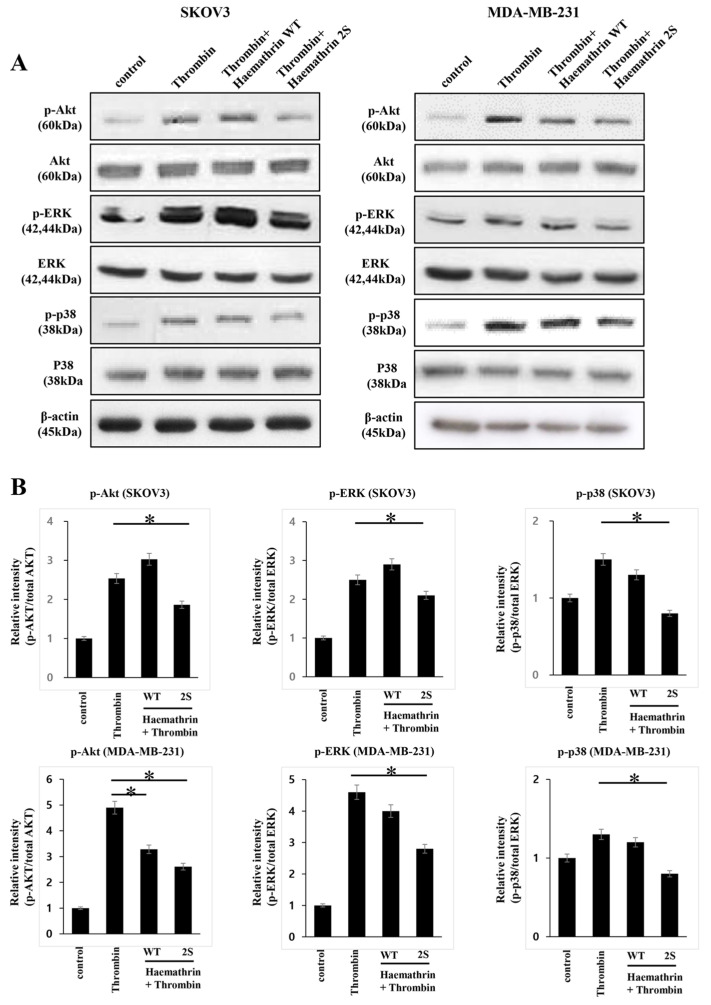
Effects of haemathrin WT and haemathrin 2S on signaling pathways. SKOV3 and MDA-MB-231 cells were pretreated with GST, haemathrin WT, or haemathrin 2S (10 μg/mL for 30 min) prior to treatment with thrombin (2 units/mL for 15 min). The expression levels of phosphorylated (p)-Akt, Akt, p-extracellular signal-regulated kinase (ERK), ERK, p-p38, and p38 protein in SKOV3 and MDA-MB-231 cells were analyzed using western blot. (**A**) Representative gel images are shown. (**B**) In SKOV3 cells, thrombin-induced p-Akt, p-ERK, and p-p38 expression was significantly attenuated by haemathrin 2S treatment but not haemathrin WT. In MDA-MB-231 cells, thrombin-induced p-Akt expression was significantly inhibited by both haemathrin 2S and WT treatment. Thrombin-induced p-ERK and p-p38 expression was significantly inhibited only by haemathrin 2S treatment. The protein levels were normalized to the level of β-actin in the same sample. Results are presented as the mean relative density ratio ± standard deviation of three independent experiments performed in triplicate (* *p* < 0.05). Haemathrin WT, haemathrin wild-type proteins; haemathrin 2S, haemathrin 2 sulfation proteins.

**Figure 4 ijms-25-11822-f004:**
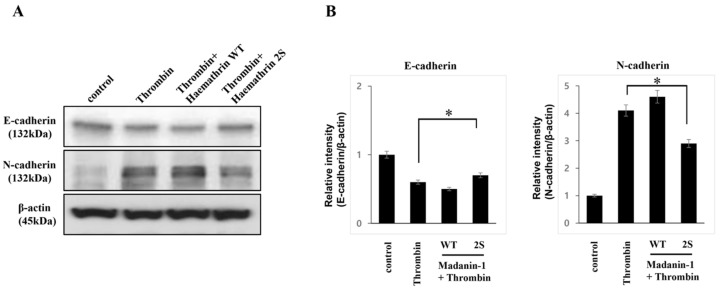
Effects of haemathrin WT and haemathrin 2S on the expression of E-cadherin, N-cadherin, and vimentin in MDA-MB-231 cells. MDA-MB-231 cells were pretreated with GST, haemathrin WT, or haemathrin 2S (10 μg/mL for 30 min) prior to treatment with thrombin (2 units/mL for 24 h). The expression levels of E-cadherin, N-cadherin, and vimentin protein were analyzed in MDA-MB-231 cells by western blot. (**A**) Representative gel images are shown. (**B**) Haemathrin 2S significantly increased E-cadherin expression, which was suppressed by thrombin treatment. The expression of N-cadherin induced by thrombin was decreased with statistical significance by haemathrin 2S. The protein levels were normalized to the level of β-actin in the same sample. Results are presented as the mean relative density ratio ± standard deviation of three independent experiments performed in triplicate (* *p* < 0.05). Haemathrin WT, haemathrin wild-type proteins; haemathrin 2S, haemathrin 2 sulfation proteins.

**Figure 5 ijms-25-11822-f005:**
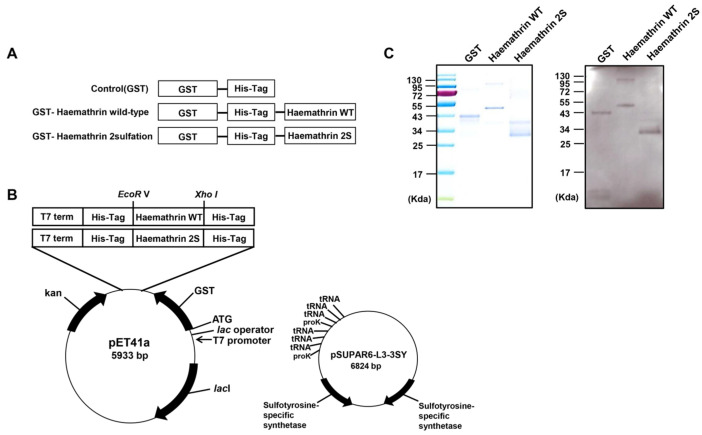
Cloning and purification of haemathrin WT and haemathrin 2S proteins. (**A**) Illustration of the expressed control, haemathrin WT, and haemathrin 2S proteins. Haemathrin was cloned into the *EcoRV* and *XhoI* restriction sites of the pET-41a vector. (**B**) Along with the pET41a plasmid encoding haemathrin WT and haemathrin 2S, the pSUPAR6-L3-3SY plasmid encoding the components necessary for translational incorporation of sulfotyrosine in response to the TAG codon was used. (**C**) Purified fusion proteins were detected via Coomassie Brilliant Blue staining after 10% SDS-PAGE (**left side**) and western blotting (**right side**). GST, Haemathrin WT, haemathrin wild-type proteins; haemathrin 2S, haemathrin 2 sulfation proteins.

**Table 1 ijms-25-11822-t001:** Sequences of haemathrin 2 nucleotides and amino acids used in this study.

Name	Type	Sequence
Haemathrin2 WT	Nucleotide	5′-ATG AAG CAC TTC GCA ATT TTT ATT CTT GCT GTT GTG GCC AGT GCC GTG GTG ATG GCA TAC CCG GAG CGC GAT TCA GCA AAT CGT GGC AGC CAA GAG AAA GAG CGC GCT CTG CTT GTT AAA GTA CAA GAA CGT TCT AGC CAA GAT GAC **TAC** GAT GAA **TAT** GAT GCA GAT GAG ACC ACT CTT TCT CCG GAT CCA GAT GCA CCG ACT GCC CGT CCA CGC CTC GGT CGT AAG AAT GCT TGA-3′
Haemathrin2 2S	Nucleotide	5′-ATG AAG CAC TTC GCA ATT TTT ATT CTT GCT GTT GTG GCC AGT GCC GTG GTG ATG GCA TAC CCG GAG CGC GAT TCA GCA AAT CGT GGC AGC CAA GAG AAA GAG CGC GCT CTG CTT GTT AAA GTA CAA GAA CGT TCT AGC CAA GAT GAC **TAG** GAT GAA **TAG** GAT GCA GAT GAG ACC ACT CTT TCT CCG GAT CCA GAT GCA CCG ACT GCC CGT CCA CGC CTC GGT CGT AAG AAT GCT TGA-3′
Haemathrin2 WT	Amino acid	5′-MKHFAIFILAVVASAVVMAYPERDSANRGSQEKERALLVKVQ ERSSQDDYDEYDADETTLSPDPDAPTARPRLGRKNA-3′
Haemathrin2 2S	Amino acid	5′-MKHFAIFILAVVASAVVMAYPERDSANRGSQEKERALLVKVQERSSQDD_ ^a^ DE_ ^a^ DADETTLSPDPDAPTARPRLGRKNA-3′

^a^ Sulfotyrosine site. The parts in bold indicate sulfation sites.

## Data Availability

The data presented in this study are available upon request from the corresponding author. Any requests should be addressed to Joon H. Lee (joonhlee@konyang.ac.kr) or Jin Sook Yoon (yoonjs@yuhs.ac).

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
