# Peer review of "Inhibition of Cancer Cell Migration and Invasion In Vitro by Recombinant Tyrosine-Sulfated Haemathrin, A Thrombin Inhibitor"

_ijms, 2024, doi:10.3390/ijms252111822_

Round 1

Reviewer 1 Report (Previous Reviewer 1)

Comments and Suggestions for Authors

The authors have submitted the revised version of the manuscript, in which all the concerns have been addressed adequately. Thanks

Author Response

Comment 1: The authors have submitted the revised version of the manuscript, in which all the concerns have been addressed adequately. Thanks

Response : Thank you for your comment. 

Reviewer 2 Report (Previous Reviewer 3)

Comments and Suggestions for Authors

In my opinion the article was improved and can be published after a minor revision.

  1. How did you decide to use 10 µg/ml haemathrin 2S and 2WT and in what did you dissolve them? Please add.

Answer: It was determined as the concentration referred to in the experiment in the previous paper. Proteins are dissolved in 20 mM Tris-Hcl (pH7.5), 50 mM Nacl.

-Jo G, Chae JB, Jung SA, Lyu J, Chung H, Lee JH. Sulfated CXCR3 Peptide Trap Use as a Promising Therapeutic Approach for Age-Related Macular Degeneration. Biomedicines. 2024 Jan 22;12(1):241. doi: 10.3390/biomedicines12010241. PMID: 38275412; PMCID: PMC10813770.

-Jo GH, Jung SA, Roh TH, Yoon JS, Lee JH. Inhibitory effect of recombinant tyrosine-sulfated madanin-1, a thrombin inhibitor, on the behavior of MDA-MB-231 and

SKOV3 cells in vitro. Mol Med Rep. 2024 Jul;30(1):114. doi: 10.3892/mmr.2024.13238. Epub 2024 May 17. PMID: 38757335; PMCID: PMC11099723.

Please add the information in the current article.

Author Response

Comment : In my opinion the article was improved and can be published after a minor revision. Please add the information in the current article.

Response : Thank you for your comment. We added this sentence in the manuscript (method 4.1 section) with the references. 

"The concentration of 10 µg/mL was determined based on the experiment in the previous paper [13,40], and the proteins were dissolved in 20 mM Tris-HCl (pH 7.5) and 50 mM NaCl." 

13. Jo GH, Jung SA, Roh TH, Yoon JS, Lee JH. Inhibitory effect of recombinant tyrosine-sulfated madanin-1, a thrombin inhibitor, on the behavior of MDA-MB-231 and SKOV3 cells in vitro. Mol Med Rep. 2024 Jul;30(1):114. 

40. Jo G, Chae JB, Jung SA, Lyu J, Chung H, Lee JH. Sulfated CXCR3 Peptide Trap Use as a Promising Therapeutic Approach for Age-Related Macular Degeneration. Biomedicines. 2024 Jan 22;12(1):241.

This manuscript is a resubmission of an earlier submission. The following is a list of the peer review reports and author responses from that submission.

Round 1

Reviewer 1 Report

Comments and Suggestions for Authors

Guk Heui Jo and coauthors submitted the manuscript "Inhibition of tumor cell migration and invasion in vitro by recombinant tyrosine-sulfated hematrin, a thrombin inhibitor". The manuscript is a regular article. In this manuscript, the authors provided results on the antimetastatic action of hematrin 2S, a sulfo-tyrosine-modified hematrin 2 that regulates thrombin activity. The authors' intent was to demonstrate that hematrin 2S can reduce tumor cell invasiveness and migration by interfering with the protumor activity of thrombin. The results were obtained by two assays, wound healing assay and Transwell cell invasion. The authors provided results from Western blotting demonstrating that hematrin 2S reduces thrombin-mediated activation of AKT, ERK, and p38 signaling pathways. E- and N-cadherin expression analysis has also been reported. The description of the material and methods is well done. The reported results apparently support the authors' aims.

However, three main criticisms seriously compromise the author's work.

The first is the wrong control used in the experiments, represented by tumor cells treated with thrombin alone. Since the WT hematrin and 2S hematrin used were GST-chimeric recombinant proteins, the correct control should be GST induced and purified from E. coli transformed with the same system (pET41, with (or without) pSUPAR6-L3-3SY). In this regard, it should be noted that both hematrin WT and hematrin 2S preparations were not pure, each containing more than one stained protein (at least three bands) (Figure 5C).

The second serious criticism is the lack of a stained protein band in both gel lanes in Figure 5C corresponding to the chimeric WT/GST hematrin or 2S/GST hematrin, with a calculated MW of approximately 40 KDa (based on the pET41 map, the GST/insert/6xHis-tail coding sequence is ~1100 bp). The most abundant protein shown in the gel is ~45 KDa in the hematrin WT lane and ~27 KDa in the hematrin 2S lane. Only a faint 35 KDa band is present in both lanes, with questionable significance. To demonstrate which band corresponds to the recombinant hematrin, the authors should perform a Western blot with an anti-hematrin antibody.

Finally, another important criticism is related to the amount of recombinant protein used in the experiments, whose effects, based on previous considerations, cannot be attributed solely to hematrin WT or hematrin 2S.

I regret to inform the authors that the manuscript, although interesting, in my opinion cannot be accepted for publication.

Minor

In the abstract, please change  “haemathrin 2-sulfation” (lane 14) with the correct tyrosine-sulfated haemathrin 2.

Introduction, lines 72-73, please verify the phrase “The sulfated form of madanin-1 induced a more potent migration and invasion of cancer cells in vitro than wild-type (WT) madanin-1…..” It is correct?

Author Response

Guk Heui Jo and coauthors submitted the manuscript "Inhibition of tumor cell migration and invasion in vitro by recombinant tyrosine-sulfated hematrin, a thrombin inhibitor". The manuscript is a regular article. In this manuscript, the authors provided results on the antimetastatic action of hematrin 2S, a sulfo-tyrosine-modified hematrin 2 that regulates thrombin activity. The authors' intent was to demonstrate that hematrin 2S can reduce tumor cell invasiveness and migration by interfering with the protumor activity of thrombin. The results were obtained by two assays, wound healing assay and Transwell cell invasion. The authors provided results from Western blotting demonstrating that hematrin 2S reduces thrombin-mediated activation of AKT, ERK, and p38 signaling pathways. E- and N-cadherin expression analysis has also been reported. The description of the material and methods is well done. The reported results apparently support the authors' aims.

However, three main criticisms seriously compromise the author's work.

Comment 1. The first is the wrong control used in the experiments, represented by tumor cells treated with thrombin alone. Since the WT hematrin and 2S hematrin used were GST-chimeric recombinant proteins, the correct control should be GST induced and purified from E. coli transformed with the same system (pET41, with (or without) pSUPAR6-L3-3SY). In this regard, it should be noted that both hematrin WT and hematrin 2S preparations were not pure, each containing more than one stained protein (at least three bands) (Figure 5C).

Answer: Thank you for your insightful comments. In this study, while it was stated that only thrombin was treated, GST was treated alongside it. It seems this point was not clearly described in the manuscript, and we appreciate you pointing it out. In the revised manuscript, we described in detail in the method and figure legend. Figure 1-4 were revised. We have added this information to the main text.

Regarding Figure 5C, which shows the results of Coomassie gel electrophoresis for haemathrin WT and haemathrin 2S, there are a few faint bands visible along with the main band. The reviewer pointed out that this could indicate the protein is not pure. However, we believe that the clearly visible main band is sufficiently prominent to guarantee the purity of the protein based on its size. If this remains an issue, we regret that this would be a limitation of our study.

Comment 2. The second serious criticism is the lack of a stained protein band in both gel lanes in Figure 5C corresponding to the chimeric WT/GST hematrin or 2S/GST hematrin, with a calculated MW of approximately 40 KDa (based on the pET41 map, the GST/insert/6xHis-tail coding sequence is ~1100 bp). The most abundant protein shown in the gel is ~45 KDa in the hematrin WT lane and ~27 KDa in the hematrin 2S lane. Only a faint 35 KDa band is present in both lanes, with questionable significance. To demonstrate which band corresponds to the recombinant hematrin, the authors should perform a Western blot with an anti-hematrin antibody.

Answer: Thank you for your insightful comment. We believe the actual molecular weight of a protein can vary based on its amino acid composition, which could cause deviations from this average. The main protein band shown in the gel for haemathrin WT was about ~45 kDa, and we believe this level of variation is within an acceptable margin of error. Additionally, in the haemathrin 2S lane, where haemathrin underwent tyrosine sulfation, the main band appeared around 27 kDa. Sulfation adds negative charges to a molecule, which could indeed affect its migration on a gel. The degree of influence depends on how sulfation affects the protein's overall charge-to-mass ratio, and additional negative charges due to modifications may alter migration. Recently, when we compared the sulfation of madanin with its wild-type form, a similar shift occurred (Jo GH et al. Molecular Medicine Reports 2024;30:114). It is plausible that this pattern may hold for haemathrin as well. In addition, regarding the reviewer's suggestion to perform a Western blot using an anti-haemathrin antibody to verify the actual molecular weight, we agree that this is a good idea. However, since there is currently no commercially available haemathrin antibody, we would need to generate the protein and antibody in our lab, which we have not been able to do. We acknowledge this as a limitation of our study. The addition of sulfates could shift the protein’s migration, but additional experiments would be necessary to confirm this.

Comment 3. Finally, another important criticism is related to the amount of recombinant protein used in the experiments, whose effects, based on previous considerations, cannot be attributed solely to hematrin WT or hematrin 2S.

Answer: Thank you for your comment. It was determined as the concentration referred to in the experiment in the previous paper. Proteins are dissolved in 20 mM Tris-Hcl (pH7.5), 50 mM Nacl.

-Jo G, Chae JB, Jung SA, Lyu J, Chung H, Lee JH. Sulfated CXCR3 Peptide Trap Use as a Promising Therapeutic Approach for Age-Related Macular Degeneration. Biomedicines. 2024 Jan 22;12(1):241. doi: 10.3390/biomedicines12010241. PMID: 38275412; PMCID: PMC10813770.

-Jo GH, Jung SA, Roh TH, Yoon JS, Lee JH. Inhibitory effect of recombinant tyrosine-sulfated madanin-1, a thrombin inhibitor, on the behavior of MDA-MB-231 and SKOV3 cells in vitro. Mol Med Rep. 2024 Jul;30(1):114. doi: 10.3892/mmr.2024.13238. Epub 2024 May 17. PMID: 38757335; PMCID: PMC11099723.

Comment 4. Minor

In the abstract, please change “haemathrin 2-sulfation” (lane 14) with the correct tyrosine-sulfated haemathrin 2.

Answer: corrected

Introduction, lines 72-73, please verify the phrase “The sulfated form of madanin-1 induced a more potent migration and invasion of cancer cells in vitro than wild-type (WT) madanin-1…..” It is correct?

Answer: Thank you for pointing out our error. We corrected the sentence into "The sulfated form of madanin-1 further suppressed migration and invasion of cancer cells in vitro than wild-type (WT) madanin-1"

Reviewer 2 Report

Comments and Suggestions for Authors

Jo et al. described the role of recombinant tyrosine-Sulfated Haemathrin in inhibiting cancer cell Migration and invasion in an in vitro model.

Major issues:

  • The authors used two cell lines: ovarian cancer - SKOV3 and breast cancer MDA-MB-231. Why did the authors use two different tumors and these particular cancers? It should be explained in the Introduction. The authors should address their hypothesis about why these cells are adequate for their analysis. If the primary intention was to verify whether the observed effects are universal, at least two cell lines/one tumor type, including normal cell lines, should be used in their studies.
  • To conclude the role of Akt/ERK/p38-dependent signaling pathways in the observed effects, observations should be verified additionally in the presence of specific kinase inhibitors.

Minor issue:

  • explain the acronyms whenever they appear for the first time in the text: Abstract - ERK, Introduction- Akt, MAPK etc
  • Figures 1,2 - enlarge photos, remove the graphs under pictures and make them larger
  • line 117-118 - 786% and 869%?
  • line 126 - add phosphorylation positions in pAkt, pERK, p-p38, whenever it appears in the text of figures
  • Figures 3,4 - enlarge, indicate Mw of detected proteins
  • Table 1 - indicate 5' and 3'
  • Move Figure 2 before 4.2 section
  • Provide cat. nb to Abs (4.4)
  • line 445 - remove doubled References
  • line 387 - remove extra space
  • line 284 - MDA-MB-231 not MDM-MB-231
  • Correct the language

Comments on the Quality of English Language

 Moderate editing of the English language is required.

Author Response

Jo et al. described the role of recombinant tyrosine-Sulfated Haemathrin in inhibiting cancer cell Migration and invasion in an in vitro model.

Major issues:

  1. The authors used two cell lines: ovarian cancer - SKOV3 and breast cancer MDA-MB-231. Why did the authors use two different tumors and these particular cancers? It should be explained in the Introduction. The authors should address their hypothesis about why these cells are adequate for their analysis. If the primary intention was to verify whether the observed effects are universal, at least two cell lines/one tumor type, including normal cell lines, should be used in their studies.

Answer: Thank you for your comment. In this study, we used SKOV3 (ovarian cancer) and MDA-MB-231 (breast cancer), as they are well-characterized, aggressive, and highly invasive cancer cell lines. The relevance in studying the metastatic potential and thrombin’s role in tumor invasion and migration is established in those cell lines. Thrombin is known to play a role in cancer progression, particularly in invasion and metastasis, and we believed these invasive cancer cell models were suitable for evaluating the effect of haemathrin.

As our primary focus was on assessing thrombin’s impact in aggressive cancer phenotype, we did not observe the results in normal cell lines. As the reviewer mentioned, future additional study in normal cells would be valuable to distinguish cancer-specific effects. We added above content in the introduction of revised manuscript. “We used SKOV3, ovarian cancer cell line and MDA-MB-231, breast cancer cell line which are well-characterized, aggressive and highly invasive cancer cells, as our primary focus was on assessing thrombin’s impact and the effect of thrombin inhibitor in aggressive cancer phenotype.”

  1. To conclude the role of Akt/ERK/p38-dependent signaling pathways in the observed effects, observations should be verified additionally in the presence of specific kinase inhibitors.

Answer: Thank you for your valuable comment. The current findings provided strong preliminary evidence based on the observed inhibition of thrombin induced phosphorylation of Akt, ERK, and p38 by haemathrin 2S in both SKOV3 and MDA-MB-231 cells, for the first time. As the reviewer mentioned, using specific kinase inhibitors of signal molecules would help provide a deeper mechanistic understanding of how haemathrin 2S functions in relation to these pathways. However, these further experiments were not performed in this study, however the significant and consistent effects of haemathrin 2S on inhibiting thrombin-induced migration, invasion, and phosphorylation of key signaling molecules across multiple pathways provide robust evidence of its therapeutic potential. Future studies will include the use of specific kinase inhibitors to further delineate these signaling pathways, but the strength and breadth of our current findings strongly support haemathrin 2S as a promising candidate for targeting thrombin-driven cancer progression. We have added this content in the discussion as, “Future studies will include the experiments with specific kinase inhibitor of signal molecules to further understand how haemathrin 2S functions in relation to these pathways.”

  1. Minor issue:
  • explain the acronyms whenever they appear for the first time in the text: Abstract - ERK, Introduction- Akt, MAPK etc

Answer: As per your instructions, I made the revisions and provided the abbreviations when the terms first appeared. Thank you.

  • Figures 1,2 - enlarge photos, remove the graphs under pictures and make them larger

Answer: We enlarged photographs and removed the graphs. We have additional supplementary figures for the graphs (supplementary figure 1,2)

  • line 117-118 - 786% and 869%?

Answer: To avoid confusion, we changed the sentence into “Mean density of invaded cells/ field was 15 in SKOV3 cells and 13 in MDA-MB-231 cells, which were significantly increased under thrombin stimulation into 118 in SKOV3 cells and 113 in MDA-MB-231 cells.”

  • line 126 - add phosphorylation positions in pAkt, pERK, p-p38, whenever it appears in the text of figures

Answer: As per your instructions, we added phosphorylation and removed p-.

  • Figures 3,4 - enlarge, indicate Mw of detected proteins

Answer: As per your instructions, we enlarged Fig 3 and 4, indicating Mw of proteins.

  • Table 1 - indicate 5' and 3'

Answer: We added 5’ and 3’ in the table.

  • Move Figure 2 before 4.2 section

Answer: I would like to confirm, if it is OK to change the location of Fig 2 before 4.2 (method section). I believe Fig 2 would be in the middle of result. If we move the Fig 2 in the method section, the numbers should be changed.

  • Provide cat. nb to Abs (4.4)

Answer: cat. numbers are provided.

  • line 445 - remove doubled References

Answer: corrected. Thank you.

  • line 387 - remove extra space

Answer: corrected. Thank you.

  • line 284 - MDA-MB-231 not MDM-MB-231

Answer: corrected. Thank you.

  • Correct the language

Answer: We have thoroughly edited English through editing service. (https://www.editage.co.kr)

Reviewer 3 Report

Comments and Suggestions for Authors

In the current article, the authors studied the effects of tyrosine-sulfated haemathrin, haemathrin 2S, on cancer cell behavior in comparison to haemathrin 2 WT, using SKOV3 and MDA-MB-231 cell lines. Their findings indicated that haemathrin 2S significantly impeded thrombin-induced migration and invasion in both cell lines, which was not observed with haemathrin 2WT. As a conclusion of the study, the authors underlined that haemathrin 2S represents a promising candidate for targeting thrombin in cancer therapy by suppressing cancer cell migration and invasion.

Some suggestions:

1. pg 7, lines 245-47: give please some details concerning the following sentence: “Haemathrin 2 WT pET-41a (50 μg/mL kanamycin) and haemathrin 2S pET-41a and pSUPAR6-L3-3SY (50 μg/mL kanamycin and 50 μg/mL chloramphenicol) were transformed into Escherichia coli BL21(DE3)”.

2. Figure 5 is not properly positioned in the manuscript.

3. How did you decide to use 10 µg/ml haemathrin 2S and 2WT and in what did you dissolve them? Please add.

4. Please clarify the statements:

-pg 9, lines 312-313: “…in the presence of haemathrin 312 2 WT or haemathrin 2S.  and line 322 “Equal amounts of protein…”. Which is the amount of protein?

Author Response

In the current article, the authors studied the effects of tyrosine-sulfated haemathrin, haemathrin 2S, on cancer cell behavior in comparison to haemathrin 2 WT, using SKOV3 and MDA-MB-231 cell lines. Their findings indicated that haemathrin 2S significantly impeded thrombin-induced migration and invasion in both cell lines, which was not observed with haemathrin 2WT. As a conclusion of the study, the authors underlined that haemathrin 2S represents a promising candidate for targeting thrombin in cancer therapy by suppressing cancer cell migration and invasion.

Some suggestions:

  1. pg 7, lines 245-47: give please some details concerning the following sentence: “Haemathrin 2 WT pET-41a (50 μg/mL kanamycin) and haemathrin 2S pET-41a and pSUPAR6-L3-3SY (50 μg/mL kanamycin and 50 μg/mL chloramphenicol) were transformed into Escherichia coli BL21(DE3)”.

Answer: According to the reviewer’s suggestion, details were given as follows, “To co-transform hematrin 2 WT pET-41a (50 μg/mL kanamycin) and hematrin 2S pET-41a and pSUPAR6-L3-3SY (50 μg/mL kanamycin and 50 μg/mL chloramphenicol) in BL21, this mixture is heat-shocked (usually at 42°C for 2 minutes) to facilitate the cells to absorb DNA. Then, incubate in a shaking incubator in 1 ml of LB medium for 1 hour without antibiotics. After incubation, the LB mixture is spread over LB agar plates (50 μg/mL kanamycin and 50 μg/mL chloramphenicol) and incubated for 24 hours in an incubator at 37°C. The next day, check the colony and incubate it again in LB broth to see if it contains plasmid.”

  1. Figure 5 is not properly positioned in the manuscript.

Answer: The location of Figure 5 was changed.

  1. How did you decide to use 10 µg/ml haemathrin 2S and 2WT and in what did you dissolve them? Please add.

Answer: It was determined as the concentration referred to in the experiment in the previous paper. Proteins are dissolved in 20 mM Tris-Hcl (pH7.5), 50 mM Nacl.

-Jo G, Chae JB, Jung SA, Lyu J, Chung H, Lee JH. Sulfated CXCR3 Peptide Trap Use as a Promising Therapeutic Approach for Age-Related Macular Degeneration. Biomedicines. 2024 Jan 22;12(1):241. doi: 10.3390/biomedicines12010241. PMID: 38275412; PMCID: PMC10813770.

-Jo GH, Jung SA, Roh TH, Yoon JS, Lee JH. Inhibitory effect of recombinant tyrosine-sulfated madanin-1, a thrombin inhibitor, on the behavior of MDA-MB-231 and

SKOV3 cells in vitro. Mol Med Rep. 2024 Jul;30(1):114. doi: 10.3892/mmr.2024.13238. Epub 2024 May 17. PMID: 38757335; PMCID: PMC11099723.

  1. Please clarify the statements:

-pg 9, lines 312-313: “…in the presence of haemathrin 2 WT or haemathrin 2S.”  and line 322 “Equal amounts of protein…”. Which is the amount of protein?

Answer: Thank you for your comment. We added the amount of protein in the method section, 4.4 as 30 μg/lane.

Round 2

Reviewer 1 Report

Comments and Suggestions for Authors

Although there is no commercial anti-hematrin antibody, the authors still need to provide evidence that recombinant WT and 2S hematrin proteins were produced, rather than nonspecific resin-bound proteins. The authors could at least provide a Western blotting experiment with an anti-His or anti-GST antibody. Alternatively, a Comassie gel with GST (control), GST/hematrin-WT, and GST/hematrin-2S should be shown to support effective 2S hematrin protein synthesis. I understand the authors' arguments about the net charge change related to the sulfation process, but they cannot provide clear evidence of effective recombinant protein production.

Author Response

Thank you sincerely for the detailed review comments. As per the reviewer's suggestions, conducting additional experiments, such as Western blotting or further Coomassie gel electrophoresis, would likely enhance the quality of the manuscript. We appreciate the valuable feedback. Since we need to manufacture new proteins, this process will take at least a month. If the editor and reviewer kindly allow this, we will proceed with the additional experiments as soon as possible. Thank you.

Reviewer 2 Report

Comments and Suggestions for Authors

The authors responded to all addressed issues and corrected the manuscript. Thank you.

Author Response

Thank you for your thorough review.